# Sustainable Financing for Transport Infrastructure: An Integral Approach for the Russian Federation

Alexander Chupin [1,*], Alexey Sorokin [1], Alena Veselko [1], Dmitry Morkovkin [2], Victor Ya. Pishchik [3]
and Petr V. Alekseev [3]

1   Faculty of Economics, Peoples' Friendship University of Russia (RUDN University), 6 Miklukho-Maklaya Street, 117198 Moscow, Russia
2   Departament of Economic Theory, Financial University under the Government of the Russian Federation, 49/2 Leningradsky Prospekt, 125167 Moscow, Russia
3   Department of the World Economy and Global Finance, Faculty of the International Economic Relations, Financial University under the Government of the Russian Federation, 49/2 Leningradsky Prospekt, 125167 Moscow, Russia
*   Correspondence: chupin-al@rudn.ru; Tel.: +7-977-552-5618

**Abstract:** The development of global transport infrastructure is a crucial aspect of economic growth and prosperity. In this study, the main trends in the development of global transport infrastructure are analyzed, with a focus on the state and trends in the functioning of the national transport infrastructure of the Russian Federation. By identifying key problems and determinants of financial growth, a preliminary decomposition is made to guide future development. Proceeding from the fact that the innovative development of Russia's transport infrastructure should occur through state stimulation of the development of intellectual and partnership relationships and the formation of a digital environment for managing projects for its innovative transformation at the state level, this study identifies the goals, methods, and levels of ensuring the implementation of innovative transformations in transport infrastructure. Consideration of the historical aspects of the formation of the modern economy and comprehension of the mechanisms of state management and tools to ensure innovative transformations in the sphere of transport allowed us to identify three stages (dogmatic, strategic regulation, and innovative integration) of transformation regarding the theory of the state management for the innovative development of transport infrastructure in Russia. Taking into account the content of these periods of transformation allowed us to detail the specifics of the modern stage; highlight the shortcomings; and propose theoretical and methodological foundations for revising the provisions of the state regulation of innovative development in transport infrastructure. To determine the effectiveness of the processes of the state regulation of innovation activity in the transport industry, we propose a model for assessing the effectiveness of innovation activity in the transport complex, which is presented in the form of a three-level parametric system, where the first group of indicators allows us to assess the innovation activity of economic entities in the transport industry; the second allows us to assess innovation potential; and the third allows us to assess the effectiveness of innovation. The novelty of the proposed model is the consideration of the principles of dynamism, perspective, and expediency, and its application allows us to establish the level of innovative development in transport infrastructure. An important factor in ensuring the sustainable development of a country's economy and forming its positive image in the world is its financial attractiveness. Therefore, the assessment of financial attractiveness is a priority task for the development of modern sustainable transport infrastructure in Russia.

**Keywords:** sustainability of transport infrastructure; financial development; state regulation; spatial inequality; regional–industry asymmetry; mathematical modeling; economic systems; sustainable development

## 1. Introduction

Financial attractiveness is an important factor in ensuring the sustainable development of a country's economy and forming its positive image on the world stage. The top five in the Global Financial Centres Index are Singapore, the United States, Switzerland, and Norway. The lowest places are occupied by the Central African Republic, Sudan, and North Korea. Each financial project has a specific focus and can be most effectively implemented in priority sectors. Therefore, assessing financial attractiveness is a priority in developing the sustainability of transportation infrastructure today. Scientists who consider financial attractiveness to be an objective prerequisite for financing and a quantitative characteristic expressing the volume of attracted capital investments have a different approach to the definition of this category. Thus, Marchevka A. [1] states that financial attractiveness is an integral characteristic of individual industries in terms of the efficiency of financial activity in them. Kozlova M. and Collan M. [2] support this approach and argue that financial attractiveness is a certain level of satisfaction in financial, production, organizational, or other requirements for an investor regarding a particular industry. The efficiency of financial processes in modern economic conditions is closely related to the assessment of the current state and the identification of the main trends in the development of the financial market. The process of studying the financial market includes the assessment of the financial attractiveness of infrastructure. By understanding the financial attractiveness of infrastructure projects in the transportation sector, policymakers, investors, and other stakeholders can make informed decisions that drive economic growth and promote sustainable development. The assessment of the financial attractiveness of transport infrastructure is key to ensuring sustainable economic development and attracting investment in the sector. By considering the specific focus and priorities of each project, stakeholders can maximize the positive impact of transport infrastructure investments and contribute to building a prosperous and sustainable future for a country.

It should be noted that among the main prerequisites for increasing the inflow of financing from all sources by activating the financial activity of economic entities, along with the improvement of the financial climate, is the improvement of the links connecting the entire financial mechanism and, in particular, the theoretical and methodological basis for justifying decisions on the object orientation of investments; the information base for making these decisions should contain an objective and comprehensive assessment of financial attractiveness—such as potential recipients of capital, for which, both for investors and for recipients, it is necessary to have a well-founded, comprehensive, and universal methodological approach to assessing the financial attractiveness of the object of financing; with the help of this, the former will be able to optimally place their funds, and the latter will be able to objectively assess their capabilities to attract them and achieve their goals. The results of this research allow us to conclude that many scientists whose research interests include the problem of substantiation and the development of methodological support for determining and analyzing the relationship between the sustainability of transport infrastructure and its financial attractiveness agree on the expediency of this solution on the basis of an integral assessment that combines all significant individual indicators, although there are a number of issues related to both the formation of the information base and the choice of methods in its indexing.

In addition, it is necessary to take into account the macroeconomic environment, regulatory framework, market conditions, and other external factors that may affect the financial attractiveness of an investment or financial opportunity. Such a comprehensive approach to assessing financial attractiveness will not only benefit investors and recipients of capital but will also contribute to the overall development and sustainability of the financial system.

The improvement of the theoretical and methodological framework for assessing financial attractiveness is crucial for increasing the flow of finance and revitalizing the financial activities of economic entities. Strengthening linkages within the financial mechanism and a universal approach to assessing financial attractiveness will allow investors

and recipients of capital to make informed decisions that will lead to mutual benefit and contribute to overall economic growth and development.

The construction of an integral, generalizing assessment of the relationship between the sustainability of transport infrastructure and its financial attractiveness is a very difficult task, which is explained by the practical impossibility of providing a sufficiently adequate representation of its real level using any formalized methods. It is obvious that a separate integral assessment in itself has no informative value and acquires a practical value only when it becomes the basis for comparison with these or other objects, categories, phenomena, etc., which it does characterize. When it is necessary to compare or organize objects by properties that are not directly measurable, specially developed measures of properties—rankings—are most often used.

The ranking is a generalized, integral assessment indicating the position of the *j*-th unit of the population on a certain scale [3]. It is in the possibility of comparing different objects (in terms of their financial attractiveness), which are provided by ranking, that the value of the latter lies. The ranking process is solved by calculating an aggregate indicator on the basis of a set of initial attributes for the objects under study, with their subsequent ranking based on the value of this aggregate indicator.

The construction of an integral assessment of the relationship between the sustainability of transport infrastructure and its financial attractiveness is a very difficult task, which is explained by the practical impossibility of providing a sufficiently adequate representation of its real level using any formalized methods. In this paper, we propose a methodological approach to determining and analyzing this relationship on the methodological basis of factor analysis, which solves these problems by performing the normalization procedure in a classical way and using the shares of the total variance of attributes as weights. These attributes are explained and interpreted as a result of performing factor analysis procedures and overcoming the disadvantages inherent to most of the existing methods of determining integral ranking scores, which are mainly based on additive or multiplicative operations of information convolution, and their implementation cannot identify or accounting for the relationship, mutual influence, or dynamics of individual components aggregated into an integral score.

This research paper is organized as follows: in Section 2, we present a literature review. Section 3 outlines the research methods. Section 4 discusses the results of the research. Section 5 concludes the research article.

## 2. Literature Review

Scientific interest in the problem of finance attractiveness assessment makes us turn to the analysis of accumulated theoretical material. The scientific works of such scientists as Tatsuyoshi Miyakosh [4], Cecilia Grieco [5], Mohammed Yaqoot [6], Helena F. Naspolini and Ricardo Rüther [7], Thomas Prässler [8], Panagiotis (Takis) Iliopoulos [9], and Carl Remlinger [10] are widely known.

By employing network analysis involving quantitative and qualitative assessments, ref. [4] investigates two financial attractiveness factors—countries and financial brokerage centers—as dynamic forces in the development of Asian economies.

From the perspective of business models, this paper can be useful, as it proposes a framework for defining parameters and variables that can also be adopted as benchmarks of innovative ways of financing transportation infrastructure [5].

This paper presents a set of theoretical issues concerning the attractiveness of the integration processes of economic entities from the point of view of finance. To reveal the essence of financial attractiveness, the motives of integration are investigated, and it is proved that all of them explicitly or implicitly have a financial nature regarding decentralized renewable energy systems [6].

The development of the global integration of the Russian national economy is associated with the need to realize the competitive advantages of Russia's geopolitical position, the industrial potential of its regions, and the logistics infrastructure of transport corridors.

The study by [11] considers the supply chain (distribution network) design model based on multifactor analysis, the methodology for justifying its configuration via cost drivers, and the stage of logistics industry infrastructure development.

When speaking about the financial stability and sustainability of socio-economic processes, both in individual regions and in Russia as a whole, one cannot ignore the importance of economic activity. Financial activity in the region is realized within the framework of economic policy, which is aimed at improving the regional economy. This is based on the formation of prospective regional economic policy, including the development and support of innovations and entrepreneurship. At the same time, the realization of financial policies should adequately meet the state of financial markets, the needs of the national economy, and the level of demand for products, as well as the structure of the market economy as a whole. The step-by-step development of economic policy allows us to develop a financial strategy for the region. It is the correctly chosen financial strategy that will allow regions to become attractive to investors and reveal their potential in the future [12].

A range of studies has explored sustainable financing for transport infrastructure in the Russian Federation. Polyakova [12,13] highlights the potential of public–private partnerships in addressing internal restrictions, while Smirnov [14,15] underscores the role of the financial system in achieving sustainable development goals. Oh [14,16] focuses on the challenges and opportunities in the urban transport sector, particularly in secondary cities, and Mnatsakanyan [17,18] emphasizes the positive correlation between transport and energy infrastructure development and regional economic growth. These studies collectively underscore the need for a comprehensive and integrated approach to sustainable financing for transport infrastructure in Russia.

In addition to all sectors of the Russian Federation, the transportation industry is a fundamental factor. This is why, when determining the indicators of the investment attractiveness of the project, it is necessary to take into account the peculiarities of this area [14]:

— The development of a region is directly dependent on the level of formation in transportation infrastructure. On this basis, state and private investors have a great interest in activating the economic potential;
— Financial risk can be high due to the large volume of financing during the life cycle of a transportation project;
— The high payback period of financial investments when studying transportation projects for the possible investment of funds.

In conditions of limited financial support, one of the promising directions of transport infrastructure development is the attraction of private partners and their opportunities. Therefore, an important problem in the development of any type of transportation is the establishment of effective interactions between state authorities, local self-governments, and businesses in the development and regulation of transport infrastructure.

## 3. Materials and Methods

In modern practice, among the methods of aggregating values [17,19] into an integral assessment, which is the basis for the further construction of rankings, the most used are the following: the method of sums, the method of arithmetic mean, the method of geometric means, the method of coefficients, the method of sum of places, and the method of distances [20–23]. The method of sums can obtain an integral rating score by summing up the ratios of all initial indicators to their base values [24], i.e., indicators standardized in a certain way:

$$B_j = \sum_{i=1}^{m} z_i^j, j = \overline{1, n} \tag{1}$$

where

$B_j$—integral ranking score of the $j$-th enterprise;

$z_i^j$—standardized value of the $i$-th indicator of the $j$-th enterprise;

*m*—number of indicators;

*n*—number of units in the ranking.

When considering theoretical aspects of aggregation methods [25], it is necessary to emphasize two key points: the issue of choosing a basic indicator in order to ensure the normalization of primary indicators concerning the issue of establishing weights for different indicators when they are rolled up.

Standardization is a necessary procedure for eliminating the influence of the dimensions of these different indicators when combining them into an integral assessment. In the most general form, standardization involves comparing the empirical values of an indicator with a certain base indicator. Statistical practice has developed many variants of standardization procedures, in particular, the classical method, the method of relations, and standardization by variation spread [26], the choice of which depends on the purpose of the study and the statistical nature of primary indicators and their socio-economic content.

Justification of the basic values of indicators is an important point in the whole computational procedure since their reality and adequacy regarding the existing economic conditions, to a certain extent, determine the value of the generalizing indicator itself.

Research on ways to establish the values of basic indicators allows us to assert that, in practice, this procedure can be carried out on the basis of the expert approach, mathematical and statistical methods, taxonomic methods, and methods of cluster analysis, as well as on the basis of solving optimization problems. Important in the design of integral indicators is the justification of the weight of individual components in their convolution, which is most often solved in two ways: either all components are given equal weights—with the disadvantage of this approach is the effect of equalizing the degree of influence that individual indicators have on the resulting one, although the existence of such a gradation in nature is objective—or the weight coefficients are established by experts, which, conversely, has a certain subjectivity in the obtained results.

The arithmetic mean method can obtain integral ranking scores via the simple averaging of the standardized values of indicators (2)—the simple arithmetic mean, by averaging and taking into account the weighting of indicators (3) or by averaging of indicators, previously aggregated by their economic content into groups, taking into account both the weighting of indicators within each group and group weights (4)—thus, the weighted arithmetic mean can be determined:

$$B_j = \frac{1}{m} \sum_{i=1}^{m} z_i^j, \ j = \overline{1, n} \tag{2}$$

$$B_j = \frac{\sum_{i=1}^{m} \left( z_i^j \delta_i \right)}{\sum_{i=1}^{m} \delta_i}, \ j = \overline{1, n} \tag{3}$$

$$B_j = \frac{\sum_{l=1}^{p} \left( \frac{\sum_{i=1}^{S} \left( z_{il}^j \delta_{il} \right)}{\sum_{i=1}^{S} \delta_{il}} * \varphi_l \right)}{\sum_{l=1}^{p} \varphi_l}, \ j = \overline{1, n} \tag{4}$$

where

$z_{il}^j$—standardized value of the *i*-th indicator of the *j*-th object, which belongs to the *l*-th group of indicators;

$\delta_i$, $\delta_{il}$, $\varphi_l$—respectively, the weighting of the *i*-th indicator, the weighting of the *i*-th indicator in the *l*-th group of indicators (intra-group), and the weighting of the *l*-th generalizing indicator or *l*-th group of indicators (inter-group);

*s*—the number of indicators in the *l*-th group;

*p* —the number of *l* groups of indicators (number of generalized indicators).

The geometric mean method involves obtaining an integral ranking score via the simple (5) or weighted (6) (7) multiplication of the original indicators and extracting the root of the appropriate degree:

$$B_j = \sqrt[m]{\prod_{i=1}^{m} z_i^j}, \, j = \overline{1,n} \tag{5}$$

$$B_j = \sqrt[\Sigma_{i=1}^{m} \delta_i]{\prod_{i=1}^{m} z_i^{j^{\delta_i}}}, \, j = \overline{1,n} \tag{6}$$

$$B_j = \sqrt[\Sigma_{l=1}^{p} \varphi_l]{\sum_{l=1}^{p} \left( \sqrt[\Sigma_{i=1}^{S} \delta_{il}]{\prod_{i=1}^{S} z_i^{j^{\delta_{il}}}} \right)^{\varphi_l}}, \, j = \overline{1,n} \tag{7}$$

In the case of obtaining an integral indicator by convoluting separate generalizing indicators calculated on the basis of primary indicators previously combined into certain groups, a combination of the above methods can also be used; i.e., when at the stage of calculating generalizing indicators, the method of the arithmetic weighted average is used, and at the stage of their subsequent convolution, the method of geometric mean (8) or vice versa (9) is used:

$$B_j = \sqrt[\Sigma_{l=1}^{p} \varphi_l]{\sum_{l=1}^{p} \left( \frac{\sum_{i=1}^{S} \left( z_{il}^j \delta_{il} \right)}{\sum_{i=1}^{S} \delta_{il}} \right)^{\varphi_l}}, \, j = \overline{1,n} \tag{8}$$

$$B_j = \frac{\sum_{l=1}^{p} \left( \sqrt[\Sigma_{i=1}^{S} \delta_{il}]{\prod_{i=1}^{S} z_{il}^{j^{\delta_{il}}}} \times \varphi_l \right)}{\sum_{l=1}^{p} \varphi_l}, \quad j = \overline{1,n} \tag{9}$$

The coefficient method repeats the procedure of the previous method except for root extraction.

The sum of places method provides an integral ranking score through the simple or weighted (10) summation of the ranks obtained for each of all original indicators:

$$B_j = \sum_{i=1}^{m} S_i^j \delta_i, \, j = \overline{1,n} \tag{10}$$

where

$S_i^j$—the rank (place) of the $j$-th enterprise based on the $i$-th indicator.

The distance method can establish the integral ranking score depending on the metric distance, usually Euclidean, between specific values of the indicators of the studied objects and a reference point in $n$-dimensional space:

$$B_j = 1 - \frac{b_{j0}}{b_0} = \frac{\sqrt{\sum_{i=1}^{m} \left( x_i^j - x_i^{em} \right)^2}}{\overline{b_0} + 2\vartheta_0} = \frac{\sqrt{\sum_{i=1}^{m} \left( x_i^j - x_i^{em} \right)^2}}{\frac{1}{n}\sum_{j=1}^{n} b_{j0} + 2\sqrt{\frac{1}{n} \sum_{j=1}^{n} \left( b_{j0} - \overline{b_0} \right)^2}}, \, j = \overline{1,n} \tag{11}$$

where

$b_{j0}$—the Euclidean distance of the $j$-th object to the reference point;

$b_0$—distance to the reference point, taking into account standard deviations;

$x_i^j$, $x_i^{em}$—respectively, the value of the $i$-th indicator at the $j$-th enterprise and at the reference object.

Thus, the integral assessment can combine one indicator and many different indicators by name, units of measurement, weighting, or other characteristics, which simplifies the assessment procedure and sometimes is the only possible option for conducting it and providing objective final conclusions.

The practical significance of calculating integral indicators as a basis for building rankings, which are an integral part of the finance process, is due to the fact that, firstly,

properly constructed indicators combine the main points of the studied aspect of the function of transport infrastructure facilities; secondly, it is always easier to analyze the change in one indicator under the influence of many different factors; and thirdly, one integral indicator, in contrast to many, even standard, indicators, provides a ranking of transport infrastructure facilities.

At the same time, the above methods of aggregating many indicators into an integral assessment are not without significant drawbacks, among which, the main ones are the following:

- The problem of selecting a baseline indicator in order to standardize the primary indicators;
- The subjectivity of the justification of the weight of individual indicators in their subjective justification of the weighting of individual indicators in the process of their minimization, which is carried out mainly by experts;
- The absence of an established institute of experts;
- Aggregation using the method of sums of indicators with different scales or different importance may lead to incorrect results;
- The zeroization of at least one of the components immediately makes the integral evaluation obtained by the coefficient method or the geometric mean method equal to zero; moreover, in the case of the application of the latter method and the extraction of the root with an even degree, the sub-root expression should satisfy the condition of non-negativity;
- It is possible that the objects whose individual values differ by an order of magnitude or more will occupy close positions in the final ranking, determined by the sum of places method and vice versa;
- The possibility of a certain distortion in the ranking results of the objects in cases of using the distance method; since it is provided by comparison with an abstract standard, the quantitative values of its attributes may have improper validity;
- The absence of the possibility of the direct construction of an aggregate indicator, as well as the possibility of a situation, in the case of non-normalized axes, where two objects that differ greatly by only one attribute will turn out to be distant from each other in Euclidean space.

Note that the first two disadvantages are inherent to all of the considered methods of integral ranking score construction, while the others are characteristic only of specific aggregation methods and can be overcome by choosing the most appropriate method for the research context, as well as by introducing certain caveats into the calculation models.

Today, to reflect and model real phenomena and processes, the natures of which are inherently multidimensional, as well as when making comparisons in a set of multidimensional objects, methods of multivariate statistical analysis are widely used that can achieve, in research, both completeness in the theoretical description of observed objects and the objectivity of subsequent conclusions. They include such statistical methods as multiple correlation and regression analysis; multidimensional scaling; the principal component method; and factor, discriminant, cluster, and taxonometric analyses. The use of multivariate statistical analysis methods implies an appeal to the system analysis of the phenomenon under study, its main components, and their relationships. So, given the dynamism and variability of the finance environment, due to the constant changes in its main parameters, the approach to assessing the financial attractiveness of transport infrastructure should be systemic, in which the process of such an assessment is considered a set of interrelated elements, the analysis of which allows us to make a final decision based on an integral indicator.

In such circumstances, an effective tool for assessing finance attractiveness is the factor analysis method, which is widely used to study the impact of various factors on the state and dynamics of a system and open up the possibility of an integrated assessment and analysis of phenomena that inherent emergent properties.

In the broadest sense, factor analysis is a set of models and methods aimed at identifying, constructing, and analyzing internal factors based on information about their "external" manifestations. The initial prerequisite for factor analysis is the existence of a relationship between several features observed simultaneously, which implies the existence of certain causes and special conditions; i.e., it is explained by the influence of certain common hidden factors, which, in fact, can only be judged by the magnitude and interrelationships of the initially studied features. In other words, the original features correlate with each other because they are influenced by a common hidden factor, the correlation of object properties with factors is primary, and the correlation of features with each other is secondary.

Factor analysis is always based on the assumption that the variables being measured are only a form of a manifestation of a variable that remains "in the background" and cannot be directly measured, the so-called latent variable, a generalizing characteristic or factor that, in many cases, has a real-world equivalent [27,28].

The purpose of factor analysis is to identify—using an appropriate calculation procedure, among a large number of directly observed features,—hypothetical values that would have a meaningful interpretation, to describe the object of study to the fullest extent possible, to reproduce the original data obtained as a result of observation and correlation between them in the simplest but most accurate way, and to explain the internal regularities that objectively exist between the original features.

Thus, factors are understood as hypothetical, indirectly measurable indicators that are, to some extent, related to the measured elementary features that are established as a result of generalizing the latter; that act as integrated characteristics or signs but of a higher level; and that were previously impossible to observe.

The linear factor analysis model is the best-known representative of the class of latent variable statistical models—models that describe and explain observed variables and the relationships between them by, depending on unobserved characteristics, mathematically constructing a set of theoretical latent variables and the function derived from them, which would sufficiently approximate the empirical values of the original variables and the relationships between them:

$$x_{ij} = f(\omega_1, \omega_2, \ldots, \omega_r) \text{ or } X = f(\omega), i = \overline{1, m}; \ j = \overline{1, n} \tag{12}$$

where

$X$—a set (a matrix of size $m \times n$) of input data, $x_{ij}$, values for $i$ variables (traits) in $j$ individuals;

$\omega$—a set (row matrix of size r) of factors (latent variables);

$m$—number of variables (features);

$n$—number of individuals;

$r$—the number of factors.

The concept of latency, which means the implicitness of characteristics that are revealed by means of factor analysis methods, is the key concept; hence, we assume that the financial attractiveness of transport infrastructure is a latent generalizing feature that manifests itself on the surface of phenomena in the form of a set of initial signs. At the same time, the fundamental difference between factor analysis and other methods of statistical analysis is that, here, economic objects, phenomena, processes, etc., are considered by taking into account not one or two but a certain set of features, which contributes to the completeness of their description and increases the objectivity of the conclusions drawn.

The projections of the values of the new latent variables that will be identified as a result of the implementation of the factor analysis procedure will characterize the level of financial attractiveness in a concise economic form and will form the basis for building a rating according to this criterion.

In general, the goal of any factor analysis method is to represent the value, $z_{ij}$, as a linear combination of $r$ hypothetical variables or factors—the above equality expresses the basic model of factor analysis:

$$z_{ij} = a_{i1}u_{1j} + a_{i2}u_{2j} + \cdots + a_{ir}u_{rj} = \sum_{\gamma=1}^{r} a_{i\gamma}u_{\gamma j}, \, i = \overline{1,m}; \, j = \overline{1,n}$$

or

$$\begin{bmatrix} z_{11} & z_{12} & \cdots & z_{1n} \\ z_{21} & z_{21} & \cdots & z_{2n} \\ \cdots & \cdots & z_{ij} & \cdots \\ z_{m1} & z_{m2} & \cdots & z_{mn} \end{bmatrix} = \begin{bmatrix} a_{11} & a_{12} & \cdots & a_{1r} \\ a_{21} & a_{21} & \cdots & a_{2r} \\ \cdots & \cdots & z_{i\gamma} & \cdots \\ a_{m1} & a_{m2} & \cdots & a_{mr} \end{bmatrix} \times \begin{bmatrix} u_{11} & u_{12} & \cdots & u_{1n} \\ u_{21} & u_{21} & \cdots & u_{2n} \\ \cdots & \cdots & u_{\gamma j} & \cdots \\ u_{r1} & u_{r2} & \cdots & u_{rn} \end{bmatrix} \tag{13}$$

$$Z = A \times U$$

where

$Z$—a matrix of order, $m \times n$, of standardized output data, $z_{ij}$;

$A$—factor mapping is a matrix of order, $r \times m$, the elements of which are the factor loads, $a_{ij}$

$U$—a matrix of order, $r \times n$, for values of all factors for all individuals, $u_{\gamma j}$.

Each column of matrix $A$ represents a factor that, according to the basic assumption of factor analysis, is behind the observed correlations, causally determining the latter factor loadings, which show which factor is associated with which variables and their numerical values, which are the coefficients of the regression of factors in variables, providing a numerically formal, i.e., mathematical, explanation of the observed correlation coefficients.

Pearson's pairwise correlation coefficient between the $i$-th and $\gamma$-th indicators is calculated by the following known formula:

$$r_{i\gamma} = \frac{\sum_{j=1}^{n} \left(x_{ij} - \overline{x_i}\right)\left(x_{lj} - \overline{x_l}\right)}{\sqrt{\sum_{j=1}^{n}\left(x_{ij} - \overline{x_i}\right)^2 \times \left(x_{lj} - \overline{x_l}\right)^2}} = \frac{\vartheta_{il}}{\vartheta_i * \vartheta_l} = \frac{\sum_{j=1}^{n}\left(z_{ij}z_{lj}\right)}{\sqrt{\sum_{j=1}^{n}z_{ij}^2 \times \sum_{j=1}^{n}z_{lj}^2}} \vartheta_{il} = \frac{\sum_{j}^{n}\left(z_{ij}z_{lj}\right)}{n-1} \, i, l = \overline{1,m};$$

where

$\vartheta_{il}$—the covariance of the $i$-th and $l$-th indicators;

$\vartheta$—standard deviation.

## 4. Results and Discussion

Thus, we can obtain an algorithm that implements the proposed methodological approach to determining the integral ranking score of the relationship between the sustainability of transport infrastructure and its financial attractiveness based on the use of factor analysis methods and a brief description of its stages [29,30].

Stage I: The formation of matrix X of size $m \times n$, that is, the initial data—these are the financial and economic coefficients of Russia's transportation infrastructure. This forms indicators for use in the evaluation process. At the same time, the information used should meet the requirements of reliability, completeness, significance, timeliness, clarity, etc. The question of the directions is now debatable; the number and structure of indicators should form the basis for determining the financial attractiveness of the transport infrastructure.

We share the opinion of many scholars concerning the expediency of using financial and economic ratios as such indicators, as we believe that financial and economic conditions are a direct reflection of the efficiency of transport infrastructure, which is the most concentrated, but at the same time, it is best to characterize it, although its assessment requires a limited amount of primary information available to external users. It should be noted that the proposed set of indicators can be changed at the request of the analyst, depending on their preferences or available information available without the need for a new expert evaluation procedure.

Stage II: The standardization of initial features, the procedure of which, in the context of factor analysis, as a rule, is implemented in a classical way.

Stage III—The calculation of the correlation matrix, R, of size $m \times m$. Because of the preliminary standardization of the initial signs, the correlation matrix completely coincides with the covariance matrix, which removes the problem of choosing between them when applying factor analysis methods. At this stage, it is advisable to test the significance of the entire correlation matrix, for example, by using $X^2$—Wilks' criterion.

Stage IV: Solving the problem of commonality—We can build a reduced correlation matrix, $R_h$, by replacing the diagonal elements of the correlation matrix with a priori estimates of commonalities—shares of unit variances caused by factors common to several variables.

Formally, commonality is defined as the sum of the squares of the loads of common factors.

The loads of common and characteristic factors are connected by a certain ratio through the total variance of the traits, which, subject to their preliminary standardization, is equal to one; namely, the total variance of a trait consists of two parts—the variance caused by the presence of common factors and the variance caused by the variation in the characteristic factor.

Stage V: Solving the factor problem—finding the mapping matrix, A—establishing the number of factors necessary to explain the correlation relationship between the initial indicators, and calculation of factor loadings.

Stage VI—Solving the rotation problem—finding the factor matrix, V, that corresponds to the simple structure principle—the requirement to provide the most simple explanation for the set of variables under study in terms of factors.

Rotation is necessary because the factoring problem has no unambiguous solution when solved algebraically—there is an infinitely large number of mapping matrices that will reproduce the reduced matrix equally well, but only when restrictions are introduced—furthermore, rotation criteria define, from a certain point of view, the optimal position of the coordinate system in the space of general factors in order to represent the configuration of vector indicators, the position of which is determined by the values of factor loadings.

Stage VII—The interpretation of the obtained factors. The task of recognizing the selected factors and defining names for them is solved subjectively on the basis of the weight of coefficients from the mapping matrix.

Stage VIII—The estimation of the values of the selected common factors for each $j$-th object—finding the matrix of the estimates of the factor values, $\hat{Q}$, with the subsequent assessment of the accuracy of the obtained values using the coefficients of multiple determination.

## 5. Conclusions

A range of studies have explored the issue of sustainable financing for transport infrastructure in the Russian Federation. Kulichenkov [31] and Ikhsanova et al. [32] both highlight the increasing importance of public–private partnerships (PPPs) and extrabudgetary funds in this context. Kulichenkov [31] specifically emphasizes the role of PPPs in attracting investment and achieving national transport objectives, while Ikhsanova [32] underscores the declining significance of budget funding and the growing importance of private and foreign investments. Adamaitis [33] further delves into the potential for PPP development in different regions of Russia, proposing an integral index that considers institutional, budgetary, and socio-economic indicators. Misanova [34] adds to this discussion by emphasizing the need for integrated innovative development in the transport infrastructure sector, particularly in the face of market constraints. These studies collectively underscore the importance of PPPs and extrabudgetary funds in financing transport infrastructure in Russia and the need for a comprehensive approach that considers regional disparities and market challenges.

However, in the present day, there are several methods of solving this problem in order to apply the integral approach of transport infrastructure financing, among which, the most widely used are as follows: the principal component method; the principal axes

method; the least squares method, varieties of which include the principal axes method with iterations of the generalization and minimum residuals method; the centroid method; the maximum likelihood method; canonical factor analysis; alpha factor analysis; and pattern analysis. Computational procedures reflecting the content of these methods are implemented in standard programs included in most statistical data analysis packages. In our opinion, it is advisable to use the principal component method to obtain the initial factor structure given its fundamental importance in multivariate factor analysis. Among the criteria for estimating the number of selected factors, the most frequently used are significance criteria related to the maximum likelihood and the least squares methods; various rules formulated in terms of eigenvalues; the criterion based on the value of the factor variance shares; dropout criteria; interpretation and invariance criteria, in particular, the Kaiser, Cattell, Horn, Lawley, and Maxwell criteria; and other criteria and possible combinations thereof.

This analysis of existing methodological approaches to building a system for rating transport infrastructure-based finance attractiveness has shown that, in this area of research, there have already been attempts to use factor analysis methods to solve this problem. However, unlike the proposals set out in those, the originality of this methodological approach lies in recommendations for building this rating system using (as a ranking) not only individual estimates of the values of only the first selected factors—which accounts for the maximum of the explained variance—but also the integral indicator obtained via the weighted summation of individual estimates of the values of all selected and interpreted factors, where the weights are the contributions of each factor explanation of the total variance (sometimes, the total commonality) of the original features, i.e., the share of variance factors in the total variance.

The selected factors, based on their essential interpretations, correspond to concepts that generalize related indicators of the financial condition of an object that is representative of its solvency, liquidity, financial stability, business activity, etc. Therefore, a rating based only on individual estimates of the values of the first selected factor for each object will not have sufficient objectivity because failure to take into account the values of other selected factors during ranking may lead to the classification of a particular object as financially unattractive only because it has the same value as a certain group of financial indicators that "load" this factor unsatisfactorily. It is worth noting that the results of this study of the most common approaches to the construction of integral indicators show that the quality of such assessments significantly depends on the methodology chosen for generalizing the input information. In this case, two main tasks arise: the justification of the values of basic indicators in order to ensure the normalization of primary indicators; and the objective determination of the weights of different indicators when they are combined into an integral indicator.

The proposed methodological approach to assessing the relationship between the sustainability of transport infrastructure and its financial attractiveness based on the methodological basis of factor analysis, which is presented here as a latent indicator, solves these problems by performing the normalization procedure in a classical way and using, as weights, the shares of the total variance of its attributes. These attributes are explained by the relevant factors, which are identified and interpreted as a result of performing factor analysis procedures, and overcome the disadvantages inherent in most existing integral ranking score methods, which are based mainly on additive or multiplicative operations of information convolution, as their implementation cannot identify or account for the relationships, mutual influence, or dynamics of individual components aggregated into an integral score.

**Author Contributions:** Conceptualization, A.C. and D.M.; methodology, A.S. and P.V.A.; formal analysis, A.C., D.M., V.Y.P. and P.V.A.; investigation, D.M.; resources, A.C., D.M. and A.S.; data curation, A.C., D.M., A.V. and A.S.; writing—original draft preparation, A.C., D.M. and A.S.; writing—review and editing, A.C., D.M. and A.S.; project administration, A.C. and V.Y.P.; fund-

ing acquisition, D.M., A.S. and P.V.A. All authors have read and agreed to the published version of the manuscript.

**Funding:** This research was financially supported by a grant from the Russian Science Foundation, No. 23-41-10001: "Mathematical models and computer technologies of scheduling production and power generation in conditions economic uncertainty", https://rscf.ru/project/23-41-10001/, accessed on 11 December 2023.

**Institutional Review Board Statement:** Not applicable.

**Informed Consent Statement:** Not applicable.

**Data Availability Statement:** The data presented in this study are available upon request from the corresponding authors.

**Conflicts of Interest:** The authors declare no conflicts of interest. The funders had no role in the design of the study; in the collection, analyses, or interpretation of the data; in the writing of the manuscript; or in the decision to publish the results.

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
