# Peer review of "Sustainable Financing for Transport Infrastructure: An Integral Approach for the Russian Federation"

_sustainability, doi:10.3390/su16083108_

Round 1

Reviewer 1 Report

Comments and Suggestions for Authors

The subject of the article is timely and is interesting for research. It is in line with the aim and scope of the special issue Selected Problems Arising in the Development of Sustainable Transport Systems. However, the manuscript should be revised consistently to ensure clarity and academic rigor.

The abstract is not well structured. It is recommended to adopt the following standard abstract structure: Background, Purpose, Methodology, Results/Conclusions, Contributions/implications. Financial attractiveness, which is mainly studied in the article, is not captured in the title. Moreover, the paper talks about the infrastructure in Russia, without using specific data and without specifying this in the title, abstract, introduction. It is not clear whether the approach is general or specific.

The Introduction section is quite general. Besides the general background, it should be included explicitly the literature gap, short information on the method. The Literature review is rather superficial. The literature gap is not identified, and the authors' contributions should be highlighted.

The paper uses a rather modest methodology, and the authors fail to clearly link title and method. The stages proposed for the methodology are presented in the results and discussions section, which remains practically without content.

The Conclusion section is general; detailed descriptions and interpretations are required. More evidence is needed to reach the conclusion and implications. There are no comparisons/references to other results from the literature and the positioning of the paper in relation to them.

Also, the quality of English can be improved.

Comments on the Quality of English Language

The quality of English can be improved.

Author Response

For research article «Sustainable Financing For Transport Infrastructure: An Integral Approach Of The Russian Federation»

Response to Reviewer 1 Comments

Comments 1: The abstract is not well structured. It is recommended to adopt the following standard abstract structure: Background, Purpose, Methodology, Results/Conclusions, Contributions/implications. Financial attractiveness, which is mainly studied in the article, is not captured in the title. Moreover, the paper talks about the infrastructure in Russia, without using specific data and without specifying this in the title, abstract, introduction. It is not clear whether the approach is general or specific.

Response 1: We agree with the comment. We have corrected the title of the manuscript. It now reads as follows: Sustainable Financing For Transport Infrastructure: An Integral Approach Of The Russian Federation. And we have corrected the abstract.

Comments 2: The Introduction section is quite general. Besides the general background, it should be included explicitly the literature gap, short information on the method. The Literature review is rather superficial. The literature gap is not identified, and the authors' contributions should be highlighted.

Response 2: We agree with the comment. We've adjusted the «Introduction» section and «Literature review». We thank the reviewer for helping us to improve our research.

Comments 3: The Conclusion section is general; detailed descriptions and interpretations are required. More evidence is needed to reach the conclusion and implications. There are no comparisons/references to other results from the literature and the positioning of the paper in relation to them.

Response 3: We agree that the Conclusion section is general in character. We compared our study with other findings and reflected them in the conclusion in the first paragraph.

Reviewer 2 Report

Comments and Suggestions for Authors

The paper covers the sustainability of transport infrastructure and financing. It provides a theoretical framework of references discussing several methods of aggregation of values such as the method of the sum, arithmetic mean, geometric mean, coefficient method of the sum of places and the method of the distances.

The topic is interesting but I have a few comments. A substantial part of the results section would appear to be more correctly placed in the methodology. It is, as far as I understand, a description of a potential framework.

Some of the conclusions need to be more supported. For example the sentence “In our opinion, in order to obtain the initial factor structure, it is advisable to use the method of principal components” requires more explanation. i.e. explaining why?

Also, there are no examples in the paper of the application of this framework.

I would suggest rearranging a bit the structure of the paper to clearly show: 1) what’s the proposed approach, 2) how to apply it, 3) some examples, 4) results, 5) conclusions  

I think that the authors have some good ideas and proposal but the article need to have a clearer structure

Minor comments

1)       The number of references is a bit low for a complex as complex as the one mentioned by the authors

2)       The presentation of some of text could be improved a bit.

Comments on the Quality of English Language

 Moderate language polishing suggested

Author Response

For research article «Sustainable Financing For Transport Infrastructure: An Integral Approach Of The Russian Federation»

Response to Reviewer 2 Comments

Comments 1: The topic is interesting but I have a few comments. A substantial part of the results section would appear to be more correctly placed in the methodology. It is, as far as I understand, a description of a potential framework.

Response 1: Yes, thank you for your comment. We absolutely agree with you

Comments 2: Some of the conclusions need to be more supported. For example the sentence “In our opinion, in order to obtain the initial factor structure, it is advisable to use the method of principal components” requires more explanation. i.e. explaining why?

Response 2: We agree with the comment. Principal Component Analysis (PCA) is one of the most widely used methods for data compression and multivariate data analysis. In the context of researching sustainable financing for transportation infrastructure development, PCA can be useful for a number of purposes.

1)     PCA allows reducing the dimensionality of data by analyzing the structure of dependencies between variables and highlighting the most significant components. This reduces the amount of information and highlights the main factors affecting the sustainable financing of transportation infrastructure.

2)     PCA facilitates data visualization, which allows to better understand the relationships between different financial indicators and assess their impact on transport infrastructure development. This can be particularly useful when making strategic financing decisions.

3)     PCA can help to identify hidden patterns in the data, highlight causal relationships, and optimize financing models to improve the sustainability of transport infrastructure development.

Thus, the use of principal component method in the study of sustainable financing for transportation infrastructure development can greatly facilitate data analysis, identify key factors, and make informed decisions to improve the financial sustainability of the industry.

Comments 3: Also, there are no examples in the paper of the application of this framework.

Response 3: We agree with the comment. We added examples to the Literature Review section and the conclusion section. We have also expanded the Introduction section.

Сomments 4: I would suggest rearranging a bit the structure of the paper to clearly show: 1) what’s the proposed approach, 2) how to apply it, 3) some examples, 4) results, 5) conclusions. I think that the authors have some good ideas and proposal but the article need to have a clearer structure.

Minor comments

1) The number of references is a bit low for a complex as complex as the one mentioned by the authors

2) The presentation of some of text could be improved a bit.

Response 4: We are grateful to the reviewer for his comments on our research. We are grateful to the reviewer for his comments on our research. Your comment about changing the structure of the study was well received, but most of the authors of the paper decided to keep the structure as it was originally stated. We have also taken into account your comment about the number of references and we have reflected the references that were not visualized initially. The presentation of some texts has been improved.

Round 2

Reviewer 1 Report

Comments and Suggestions for Authors

Not all observations were taken into account for redoing the study (methodology-results-discussions).

Comments on the Quality of English Language

Acceptable.

Author Response

For research article «Sustainable Financing For Transport Infrastructure: An Integral Approach Of The Russian Federation»

Response to Reviewer 1 Comments

Comments 1: Not all observations were taken into account for redoing the study (methodology-results-discussions).

Response 1: We thank you for your comment. We consulted with the authors of the article and in a long discussion we came to the conclusion that this methodology is more appropriate for this study.

Comments 2: Comments on the Quality of English Language - Acceptable.

Response 2: We thank you for evaluating our English language at a good level.

Reviewer 2 Report

Comments and Suggestions for Authors

 The authors have successfully addressed my comments.  

Comments on the Quality of English Language

Suggest the auhtors to give the manuscript another reading. Some polishing advisable 

Author Response

For research article «Sustainable Financing For Transport Infrastructure: An Integral Approach Of The Russian Federation»

Response to Reviewer 2 Comments

Comments 1: Comments and Suggestions for Authors - The authors have successfully addressed my comments. 

Response 1: We thank you for your comments. You have helped us to improve our research
